# Does deep neuromuscular blockade provide improved perioperative outcomes in adult patients? A systematic review and meta-analysis of randomized controlled trials

**Siyuan Liu**[ID]⊕, **Bin He**⊕, **Lei Deng, Qiyan Li, Xiong Wang**[ID]*

Department of Anesthesiology, Clinical Medical College & Affiliated Hospital of Chengdu University, Chengdu, Sichuan, People's Republic of China

⊕ These authors contributed equally to this work.
* ahcduwx@163.com

**Data Availability Statement:** A pre-specified study protocol was published in the International Prospective Register of Systematic Reviews

## Abstract

Deep neuromuscular blockade provides better surgical workspace conditions in laparoscopic surgery, but it is still not clear whether it improves perioperative outcomes, not to mention its role in other types of surgeries. We performed this systematic review and meta-analysis of randomized controlled trials to investigate whether deep neuromuscular blockade versus other more superficial levels of neuromuscular blockade provides improved perioperative outcomes in adult patients in all types of surgeries. Medline, Embase, Cochrane Central Register of Controlled Trials, and Google Scholar were searched from inception to June 25, 2022. Forty studies (3271 participants) were included. Deep neuromuscular blockade was associated with an increased rate of acceptable surgical condition (relative risk [RR]: 1.19, 95% confidence interval [CI]: [1.11, 1.27]), increased surgical condition score (MD: 0.52, 95% CI: [0.37, 0.67]), decreased rate of intraoperative movement (RR: 0.19, 95% CI: [0.10, 0.33]), fewer additional measures to improve the surgical condition (RR: 0.63, 95% CI: [0.43, 0.94]), and decreased pain score at 24 h (MD: -0.42, 95% CI: [-0.74, -0.10]). There was no significant difference in the intraoperative blood loss (MD: -22.80, 95% CI: [-48.83, 3.24]), duration of surgery (MD: -0.05, 95% CI: [-2.05, 1.95]), pain score at 48 h (MD: -0.49, 95% CI: [-1.03, 0.05]), or length of stay (MD: -0.05, 95% CI: [-0.19, 0.08]). These indicate that deep neuromuscular blockade improves surgical conditions and prevents intraoperative movement, and there is no sufficient evidence that deep neuromuscular blockade is associated with intraoperative blood loss, duration of surgery, complications, postoperative pain, and length of stay. More high-quality randomized controlled trials are needed, and more attention should be given to complications and the physiological mechanism behind deep neuromuscular blockade and postoperative outcomes.

(PROSPERO) database (CRD42022301248, https://www.crd.york.ac.uk/PROSPERO/display_record.php?RecordID=301248). All relevant data are within the paper and its Supporting Information files.

**Funding:** The authors received no specific funding for this work.

**Competing interests:** The authors have declared that no competing interests exist.

## Introduction

Immobility is one of the basic clinical outcomes for drug-induced reversible general anesthesia. Neuromuscular blocking agents (NMBAs) as pharmacological adjuncts to general anesthesia have greatly facilitated advances in modern surgery and anesthesia [1, 2]. During general anesthesia, NMBAs have the potential to prevent intraoperative surgical complications caused by sudden body movement and poor surgical conditions. However, whether the evidence is sufficient is worth investigating [3].

In laparoscopic surgery, the quality of the surgical condition is determined by several modifiable factors, including the depth of neuromuscular blockade (NMB), intra-abdominal pressure, and position [4]. The depth of the NMB is monitored by train-of-four (TOF) ratio monitoring and posttetanic count (PTC) monitoring [5, 6]. Following nondepolarizing neuromuscular blocking agent administration, TOF ratio = 0.4–0.9 means minimal neuromuscular blockade, TOF ratio < 0.4 means shallow neuromuscular blockade, TOF 1–3 means moderate neuromuscular blockade (MNMB), and PTC > 0 and TOF = 0 means deep neuromuscular blockade (DNMB) [7].

Previously published meta-analyses demonstrated that DNMB improved the surgical condition, facilitated low intraoperative pneumoperitoneum pressure, and reduced postoperative pain in laparoscopic surgery [8–11]. The limitation of previous meta-analyses is that they may ignore key differences between studies, such as surgery type, anesthesia type, and different NMB depths. In addition, there were not enough participants to properly address questions through previous meta-analyses.

In recent years, DNMB has been reported in other types of surgery. In laryngeal surgery, DNMB can not only facilitate laryngoscope exposure but also reduce the incidence of intraoperative vocal cord movement and cough [12, 13]. It was reported that there was no intraoperative cough and no increased incidence of postoperative atelectasis in the DNMB group during thoracoscopic surgery [14, 15]. In orthopedic surgery, researchers found less intraoperative blood loss in the DNMB group than in the MNMB group [16], while other researchers observed a lower level of inflammatory factors in the DNMB group [17].

Although DNMB provides better surgical workspace conditions in laparoscopic surgery, it is still not clear whether it decreases the duration of surgery, complications, and length of stay, not to mention its role in other types of surgeries. We performed this systematic review and meta-analysis of RCTs to investigate whether DNMB provides improved perioperative outcomes in adult patients.

## Material and methods

### Registration

We performed this systematic review and meta-analysis following the Preferred Reporting Items for Systematic Reviews and Meta-Analyses (PRISMA) guidelines [18]. (PROSPERO registration number: CRD42022301248).

### Inclusion criteria for study selection

**Studies.** We included randomized controlled trials (RCTs) regardless of publication status, date of publication, or language. We excluded other study designs and data that were not reported in the full-text article.

**Participants.** We included studies including adult patients (more than or equal to 18 years old) receiving nondepolarizing NMBAs for general anesthesia regardless of gender, race, and American Society of Anesthesiologists (ASA) grade, nationality, and educational

background. We will not include pediatric patients, healthy volunteers, or participants not undergoing surgery.

**Interventions.** We will include studies comparing DNMB (DNMB group) versus other more superficial levels of neuromuscular blockade (non-DNMB group), with no restrictions on the dosage and type of nondepolarizing NMBAs. The different depths of the neuromuscular blockade were defined by the original authors.

**Outcomes.** Primary outcomes included acceptable surgical condition, surgical condition score, intraoperative movement, and adverse events. Secondary outcomes included additional measures to improve the surgical condition, intraoperative blood loss (mL), duration of surgery (min), pain at 24 h, pain at 48 h, and length of stay (d).

The definitions are as follows: (1) Surgical condition score: Surgical condition scales based on the subjective judgment of the surgeon were commonly used to evaluate the surgical workspace condition, including the 5-point scale (optimal = 5, good but not optimal = 4, moderate = 3, poor but not optimal = 2, poor, and unacceptable = 1) [19], and the 4-point scale (excellent = 1, good but not optimal = 2, poor but acceptable = 3, unacceptable = 4) [20]. According to the method of the published research, we converted the 4-point scale to a 5-point scale so that we could pool data in the meta-analysis [8–11]. (2) Acceptable surgical condition: Based on surgical condition scales, excellent, optimal, or good but not optimal surgical conditions are not generally thought to interfere with surgical procedures. Therefore, we defined that acceptable surgical condition includes excellent, optimal, and good but not optimal surgical conditions. (3) Intraoperative movement: body movement during surgery. (4) Adverse event: intraoperative and postoperative complications that may be associated with interventions. (5) Additional measures to improve the surgical condition: measures that can improve the surgical condition, including the use of additional NMBAs, changing body position, increasing pneumoperitoneum pressure, and switching to open surgery. (6) Intraoperative blood loss: blood loss during surgery. (7) Duration of surgery: the length of time that surgery continues. (8) Pain: score of the visual analog scale (VAS) or numerical rating scale (NRS), which is converted to a 1–10 range. (9) Length of stay: hospital stays after surgery.

## Search strategy

We searched the following databases: Medline (PubMed), Embase (Ovid), Cochrane Central Register of Controlled Trials (CENTRAL), and Google Scholar from inception to June 25, 2022. We searched the reference lists of reviews, clinical trials, books, documents, and editorials. We also searched ClinicalTrials.gov and the World Health Organization International Clinical Trials Registry Platform to identify ongoing or unpublished eligible trials. We used various combinations of the following search terms: 'neuromuscular blockade', 'neuromusc*', 'Depth', 'Deep', 'profound', 'intense', 'extreme', 'moderate', 'middle', 'medium', 'middle', 'shallow', and 'low' (S1 Table).

## Study selection

Retrieved studies were imported into Endnote X9 software, and duplicates were excluded. Two reviewers (LSY and HB) independently screened the titles and abstracts to identify potentially relevant studies. Then, two reviewers (LSY and HB) independently screened the full articles to confirm the eligibility of the retrieved studies. Any disagreement was resolved by consensus and consultation with a third reviewer (WX).

### Data extraction

Two reviewers (LSY and DL) independently extracted data and imported the data into RevMan 5.4.1 software. Extracted data included author, publication year, country, inclusion/exclusion criteria, participant characteristics, the number of participants, intervention details, duration of intervention, follow-up time, and outcomes.

### Risk of bias assessment

Two reviewers (LSY and DL) independently evaluated the risk of bias using the second version of the risk-of-bias tool (ROB 2) [21]. Any disagreement was resolved by consensus and consultation with a third reviewer (WX).

### Statistical analysis

We performed data analyses using Revman 5.4.1 software. We present weighted mean differences (MDs) and 95% confidence intervals (CIs) with continuous data. We presented risk ratios (RRs) and 95% CIs with dichotomous data. If only the median, 95% CIs, ranges and interquartile ranges were reported, we converted them to MDs and standard deviations (SDs). We pooled data via the random effect model with stratification by overall risk-of-bias judgment. We assessed heterogeneity through the $I^2$ value and the Chi2 test. $I^2 > 50\%$ was defined as significant heterogeneity. We considered the following categories of $I^2$ statistics: 50%-75% (may represent substantial heterogeneity) and 75%-100% (may represent considerable heterogeneity).

   We performed subgroup analyses based on the type of surgery, type of anesthesia (total intravenous anesthesia and inhalational anesthesia), and NMB depth (Depending on whether it's MNMB or shallower NMB in the control group, we divided subgroups into the DNMB vs MNMB subgroup and the DNMB vs non-MNMB subgroup). We also conducted sensitivity analyses by converting to the fixed effect model and excluding high risk-of-bias studies. In more than ten studies, we assessed reporting bias by the funnel plot. The asymmetry of the funnel plot may be associated with reporting bias. We conducted trial sequential analyses for primary outcomes using TSA 0.9.5.10 beta software. The required information size for the outcome was based on a type I error of 0.05 and a beta of 0.2.

### Assessment of quality of evidence

To assess the level of evidence, we used the Grades of Recommendation, Assessment, Development and Evaluation (GRADE) system [22]. We divided the strength of evidence into the following levels: high, moderate, low, or very low.

## Results

### Study selection

The study selection results are summarized in the PRISM flow diagram (Fig 1). The studies were published from inception to June 25, 2022. The search strategy yielded 3747 studies after duplicates were removed, of which 45 RCTs were selected for full-text assessment. Two studies were excluded because no related outcomes were reported, and 3 studies were excluded because they were not parallel RCTs. Finally, 40 RCTs met the inclusion criteria [12–17, 19, 23–55].

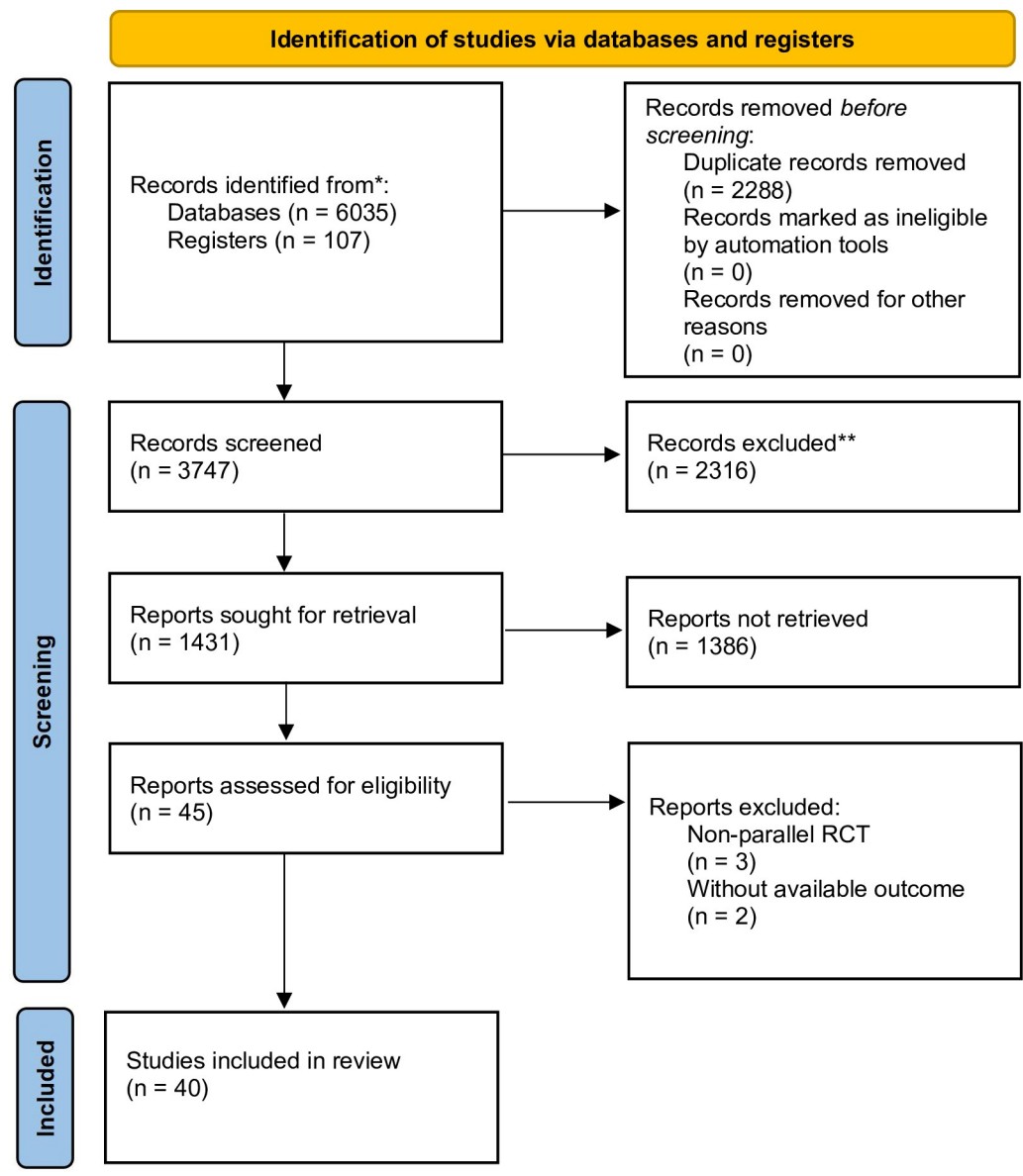

**Fig 1. Literature search and study selection process.**

## Characteristics of included studies

Forty RCTs including a total of 3271 patients were included in this systematic review [12–17, 19, 23–55]. The RCTs recruited both adult male and female adults in the American Society of Anesthesiologists classification I-IV undergoing elective noncardiac surgery under general anesthesia. The sample size ranged from 24 to 350.

The type of surgeries varied, including laparoscopic surgery [19, 23–27, 29–31, 33–38, 40, 42, 43, 45–47, 49–55], laparotomy [46], orthopedic surgery [16, 17, 28, 48], thoracoscopic surgery [14, 15], laryngeal surgery [12, 13, 41], urological surgery [39], and interventional radiography surgery [32, 44]. Except for 8 studies comparing DNMB with non-MNMB [13, 25, 29, 46–48, 50, 51], the majority of studies compared DNMB with MNMB [12, 14–17, 19, 23, 24, 26–28, 30–45, 49, 52–55]. The majority of definitions of neuromuscular blockade depth were

unanimous. Twenty studies used the TIVA technique [12–17, 19, 23, 24, 27, 30, 37, 38, 42, 46–48, 51, 52, 55], and 20 studies used the inhalational anesthesia technique [25, 26, 28, 29, 31–36, 39–41, 43–45, 49, 50, 53, 54]. The details of the study characteristics are shown in Table 1.

## Quality assessment of included studies

Seven studies were judged to be at low risk of bias [17, 23, 26, 28, 31, 40, 53], 23 studies were judged to raise some concerns [12, 15, 16, 19, 24, 29, 32, 34, 36, 38, 41–47, 49–52, 54, 55], and 10 studies were judged to be at high risk of bias [13, 14, 25, 27, 30, 33, 35, 37, 39, 48]. The details of judgments for each risk-of-bias domain are demonstrated in Fig 2.

## Primary outcomes

**Acceptable surgical condition.**   Twenty-two studies reported the rate of acceptable surgical conditions for 1984 patients [12, 15, 19, 24, 26–29, 32, 34–36, 38–42, 46, 51–53, 55]. A total of 832/993 patients in the DNMB group had an acceptable surgical condition, and 692/991 patients in the non-DNMB group had an acceptable surgical condition. The pooled data showed that patients who received DNMB were more likely to have an acceptable surgical condition than those who received shallower NMB, and the heterogeneity between studies was moderate (RR: 1.19, 95% CI: [1.11, 1.27], $p < 0.00001$, $I^2 = 55\%$) (Fig 3).

In both the subgroups receiving orthopedic surgery and laryngeal surgery, the differences between groups were not significant (1 RCT, RR: 1.06, 95% CI: [0.60, 1.89], $p = 0.84$; 2 RCTs, RR: 1.29, 95% CI: [0.92, 1.79], $p = 0.14$, $I^2 = 79\%$). In the DNMB vs non-MNMB subgroup, the difference between groups was not significant (3 RCTs, RR: 1.25, 95% CI: [0.88, 1.78], $p = 0.21$, $I^2 = 72\%$). In the other subgroup analyses, the results were stable (S1 Fig).

**Surgical condition score.**   Twenty-eight studies reported the surgical condition score for 2276 patients: 7/28 used the 4-point scale, and 21/28 used the 5-point scale [12, 13, 15, 16, 19, 23, 24, 26–29, 31, 32, 34, 35, 38–40, 42–44, 46, 49, 51–55]. The pooled mean surgical condition was 4.4±0.8 in the DNMB group and 4.0±1.1 in the non-DNMB group. The pooled data showed that patients who received DNMB had significantly improved surgical condition scores compared to those who received shallower NMB, and the heterogeneity was considerable (MD: 0.52, 95% CI: [0.37, 0.67], $p < 0.00001$, $I^2 = 85\%$) (Fig 4).

In the subgroup receiving orthopedic surgery, the differences between groups were not significant (2 RCTs, MD: 0.28, 95% CI: [-0.30, 0.87], $p = 0.34$, $I^2 = 82\%$). In the other subgroup analyses, the results were stable (S2 Fig).

**Intraoperative movement.**   Eleven studies reported the incidence of intraoperative movement for 883 patients [12, 13, 25, 36–39, 41, 46, 48, 55]. A total of 24/445 patients in the DNMB group had intraoperative movements, and 122/438 in the non-DNMB group. The pooled data showed that the DNMB group had a lower incidence of intraoperative movement than the non-DNMB group, and the heterogeneity might not be important (RR: 0.19, 95% CI: [0.10, 0.33], $p < 0.00001$, $I^2 = 24\%$) (Fig 5).

In the subgroup receiving orthopedic surgery, the differences between groups were not significant (1 RCT, RR: 0.19, 95% CI: [0.01, 3.77], $p = 0.27$). In the other subgroup analyses, the results were stable (S3 Fig).

**Adverse events.**   Twenty-one studies reported specific perioperative adverse events, which included but were not limited to vascular and organ injury [39, 44, 49], respiratory complications [14, 23, 34, 35, 39, 40], cardiovascular complications, digestive complications [34], postoperative nausea and vomiting [17, 30, 31, 33, 38, 40, 41, 44, 48, 50, 52, 53], incision complications [46], postoperative delirium [17], and shoulder pain [27, 38, 47]. The sample size was small, and the adverse events varied. Additionally, it was uncertain whether these

**Table 1. The details of the included studies.**

| Study | N | Country | ASA [a] | Surgery | Experimental | Control | Anesthesia | Funding source |
|---|---|---|---|---|---|---|---|---|
| Baete 2017 | 60 | Belgium | I-III | laparoscopic surgery | DNMB [b] (PTC > 1 and TOF = 0) | MNMB [c] (TOF 1–3) | TIVA [e] | Merck, Sharpe & Dohme |
| Barrio 2017 | 60 | Spain | I-II | laparoscopic surgery | DNMB (PTC < 5 and TOF = 0) | MNMB (TOF 1–3) | TIVA | None |
| Blobner 2015 | 50 | Germany | I-III | laparoscopic surgery | DNMB (PTC < 2) | Non-MNMB [d] (no NMB) | IA [f] | None |
| Boggett 2020 | 350 | Australia | I-III | laparoscopic surgery | DNMB (PTC < 3) | MNMB (TOF > 2) | IA | Merck, Sharpe & Dohme |
| Brunschot 2018 | 34 | Netherlands | No details | laparoscopic surgery | DNMB (PTC 1–5) | MNMB (TOF 0–1) | IA | Merck, Sharpe & Dohme |
| Choi 2019 | 100 | Korea | I-III | laparoscopic surgery | DNMB (PTC 1–2) | MNMB (TOF 1–2) | TIVA | Merck, Sharpe & Dohme |
| Curry 2020 | 116 | Portland | I-III | Orthopedic surgery | DNMB (PTC 1–2 and TOF = 0) | MNMB (TOF 1–2) | IA | Departmental funds |
| Dubois 2014 | 100 | Belgium | I-II | laparoscopic surgery | DNMB (TOF < 2) | Non-MNMB (spontaneous recovery after intubation) | IA | Departmental funds |
| Gu 2021 | 91 | China | I-II | laparoscopic surgery | DNMB (PTC 1–2) | MNMB (TOF 1–2) | TIVA | Government funds |
| Honing 2021 | 98 | Netherlands | I-III | laparoscopic surgery | DNMB (PTC 1–2) | MNMB (TOF 1–2) | IA | Merck, Sharpe & Dohme |
| Kang 2020 | 88 | Korea | I-III | Orthopedic surgery | DNMB (PTC < 3 and TOF = 0) | MNMB (TOF 1–2) | TIVA | Government funds |
| Kim 2015 | 72 | Korea | I-III | Laryngeal surgery | DNMB (PTC 1–2) | MNMB (TOF 1–2) | TIVA | None |
| Kim 2016 | 61 | Korea | I-III | laparoscopic surgery | DNMB (PTC 1–2) | MNMB (TOF 1–2) | IA | None |
| Kim 2019 | 56 | Korea | I-III | laparoscopic surgery | DNMB (PTC 1–2) | MNMB (TOF 1–2) | IA | None |
| Kim 2020 | 57 | Korea | I-II | Interventional radiography | DNMB (PTC 1–2) | MNMB (TOF 1–2) | IA | None |
| Kim 2021 | 46 | Korea | I-II | laparoscopic surgery | DNMB (PTC 1–2) | MNMB (TOF 1–2) | IA | Departmental funds |
| Koo 2016 | 64 | Korea | I-II | laparoscopic surgery | DNMB (PTC 1–2) | MNMB (TOF 1–2) | TIVA | Departmental funds |
| Koo 2018 | 64 | Korea | I-II | laparoscopic surgery | DNMB (PTC 1–2) | MNMB (TOF 1–2) | IA | Departmental funds |
| Koo 2019 | 88 | Korea | I-II | laparoscopic surgery | DNMB (PTC 1–2) | MNMB (TOF 1–2) | TIVA | None |
| Koo and Chung 2019 | 104 | Korea | I-II | Urological surgery | DNMB (PTC = 2 and TOF = 0) | MNMB (TOF 1–2) | IA | None |
| Koo 2021 | 58 | Korea | I-II | laparoscopic surgery | DNMB (PTC 1–2) | MNMB (TOF 1–2) | IA | Departmental funds |
| Laosuwan 2020 | 97 | Thailand | I-II | Laryngeal surgery | DNMB (PTC 1–2) | MNMB (TOF 1–2) | IA | Merck, Sharpe & Dohme |
| Lee 2021 | 77 | Korea | I-II | laparoscopic surgery | DNMB (PTC = 1) | MNMB (TOF = 1) | TIVA | Merck, Sharpe & Dohme |
| Lee and Lee 2021 | 114 | Korea | I-III | Thoracoscopic surgery | DNMB (PTC 1–2) | MNMB (TOF 1–2) | TIVA | Merck, Sharpe & Dohme |
| Leeman 2021 | 62 | Netherlands | - | laparoscopic surgery | DNMB (PTC 1–2) | MNMB (TOF 1–2) | IA | None |
| Loh 2021 | 45 | New Zealand | I-III | Interventional radiography | DNMB (PTC 1–2) | MNMB (TOF 1–2) | IA | Departmental funds |
| Lowen 2022 | 38 | Australia | I-III | laparoscopic surgery | DNMB (PTC 0–1) | MNMB (TOF = 1) | IA | Government funds |
| Madsen 2016 | 99 | Denmark | I-II | laparoscopic surgery | DNMB (PTC 1–2) | Non-MNMB (spontaneous recovery after intubation) | TIVA | Merck, Sharpe & Dohme |
| Madsen 2017 | 128 | Denmark | I-III | Laparotomy | DNMB (PTC 1–2) | Non-MNMB (spontaneous recovery after intubation) | TIVA | Merck, Sharpe & Dohme |
| Martini 2014 | 24 | Netherlands | I-II | laparoscopic surgery | DNMB (PTC 1–2) | MNMB (TOF 1–2) | TIVA | Merck, Sharpe & Dohme |

(*Continued*)

**Table 1.** (Continued)

| Study | N | Country | ASA [a] | Surgery | Experimental | Control | Anesthesia | Funding source |
|---|---|---|---|---|---|---|---|---|
| Oh 2019 | 83 | Korea | I-II | Orthopedic surgery | DNMB (PTC 1–2 and TOF = 0) | Non-MNMB (no NMB) | TIVA | Merck, Sharpe & Dohme |
| Oh 2021 | 82 | Korea | I-III | Orthopedic surgery | DNMB (PTC 1–2) | MNMB (TOF 1–2) | TIVA | Merck, Sharpe & Dohme |
| Putz 2016 | 100 | Belgium | I-II | laparoscopic surgery | DNMB (TOF < 1) | Non-MNMB (spontaneous recovery after intubation) | IA | Departmental funds |
| Putz 2022 | 33 | Belgium | I-II | Laryngeal surgery | DNMB (TOF < 1) | Non-MNMB (no NMB) | TIVA | Departmental funds |
| Staehr-Rye 2014 | 48 | Denmark | I-III | laparoscopic surgery | DNMB (PTC 1–2) | Non-MNMB (no NMB) | TIVA | Merck, Sharpe & Dohme |
| Torensma 2016 | 100 | Netherlands | I-III | laparoscopic surgery | DNMB (PTC 2–3 and TOF = 0) | MNMB (TOF 1–2) | TIVA | Merck, Sharpe & Dohme |
| Williams III 2020 | 100 | USA | II-IV | laparoscopic surgery | DNMB (PTC 1–2) | MNMB (TOF 1–2) | IA | Merck, Sharpe & Dohme |
| Yoo 2015 | 66 | Korea | I-II | laparoscopic surgery | DNMB (PTC 1–2) | MNMB (TOF 1–2) | IA | Merck, Sharpe & Dohme |
| Zhang 2018 | 58 | China | I-II | Thoracoscopic surgery | DNMB (PTC 1–5) | MNMB (TOF 1–2) | TIVA | Departmental funds |
| Zhu 2020 | 100 | China | I-II | laparoscopic surgery | DNMB (PTC 1–2) | MNMB (TOF 1–2) | TIVA | Government funds |

Abbreviations:

[a] The American Society of Anesthesiologists;

[b] Deep neuromuscular blockade;

[c] Moderate neuromuscular blockade;

[d] Non-moderate neuromuscular blockade;

[e] Total intravenous anesthesia;

[f] Inhalational anesthesia.

adverse events were associated with NMB in clinical practice. Therefore, data synthesis was not performed.

## Secondary outcomes

**Additional measures to improve the surgical condition.** Twelve studies reported additional measures to improve the surgical condition of 867 patients [14, 15, 25, 28, 32, 33, 36–38, 45, 46, 49]. A total of 104/445 patients in the DNMB group received additional measures to improve surgical conditions, and 179/438 patients in the non-DNMB group received additional measures. According to these RCTs above, 3 measures were used to improve the surgical condition, including the use of additional NMBAs [14, 15, 25, 28, 32, 33, 36, 37, 45, 46, 49], increasing pneumoperitoneum pressure [38, 45], and switching to open surgery [14, 45, 49]. The use of additional NMBAs was the most commonly used measure and was applied to all types of surgery, and the others were only reported in the laparoscopic surgery. The pooled data showed that patients who received DNMB were less likely to receive additional measures to improve the surgical condition with considerable heterogeneity (RR: 0.60, 95% CI: [0.40, 0.89], $p = 0.0005$, $I^2 = 67\%$) (S4 Fig).

The difference between additional measures to improve surgical condition events was not significant in the orthopedic, thoracoscopic, and interventional radiography surgery subgroups (1 RCT, RR: 1.38, 95% CI: [0.60, 3.17], $p = 0.45$; 2 RCTs, RR: 0.27, 95% CI: [0.01, 7.51], $p = 0.44$, $I^2 = 83\%$; 1 RCT, RR: 0.15, 95% CI: [0.01, 2.74], $p = 0.20$). The difference was not significant in either the MNMB, non-MNMB, or inhalational anesthesia

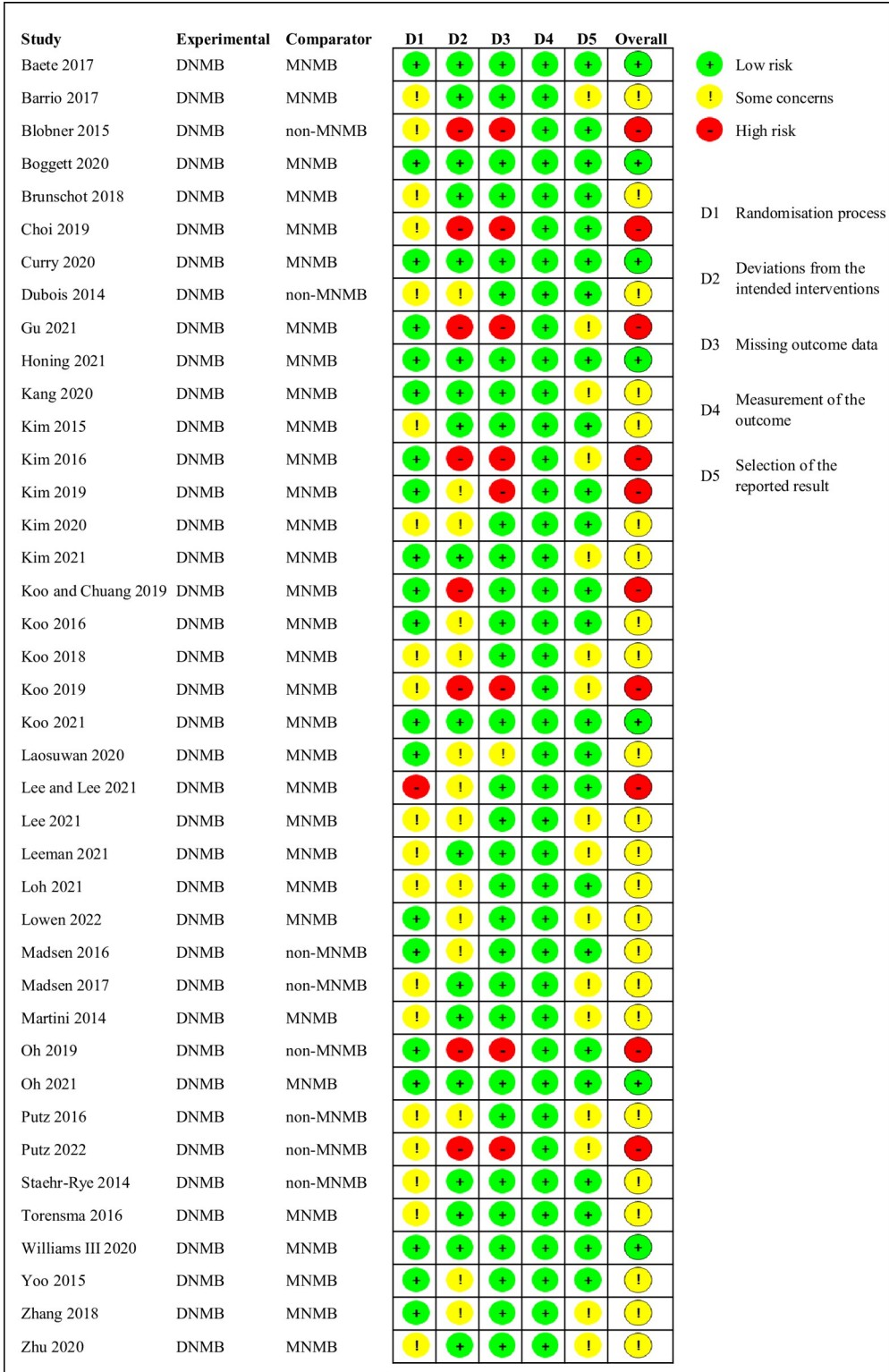

**Fig 2. Risks of bias of the included studies.**

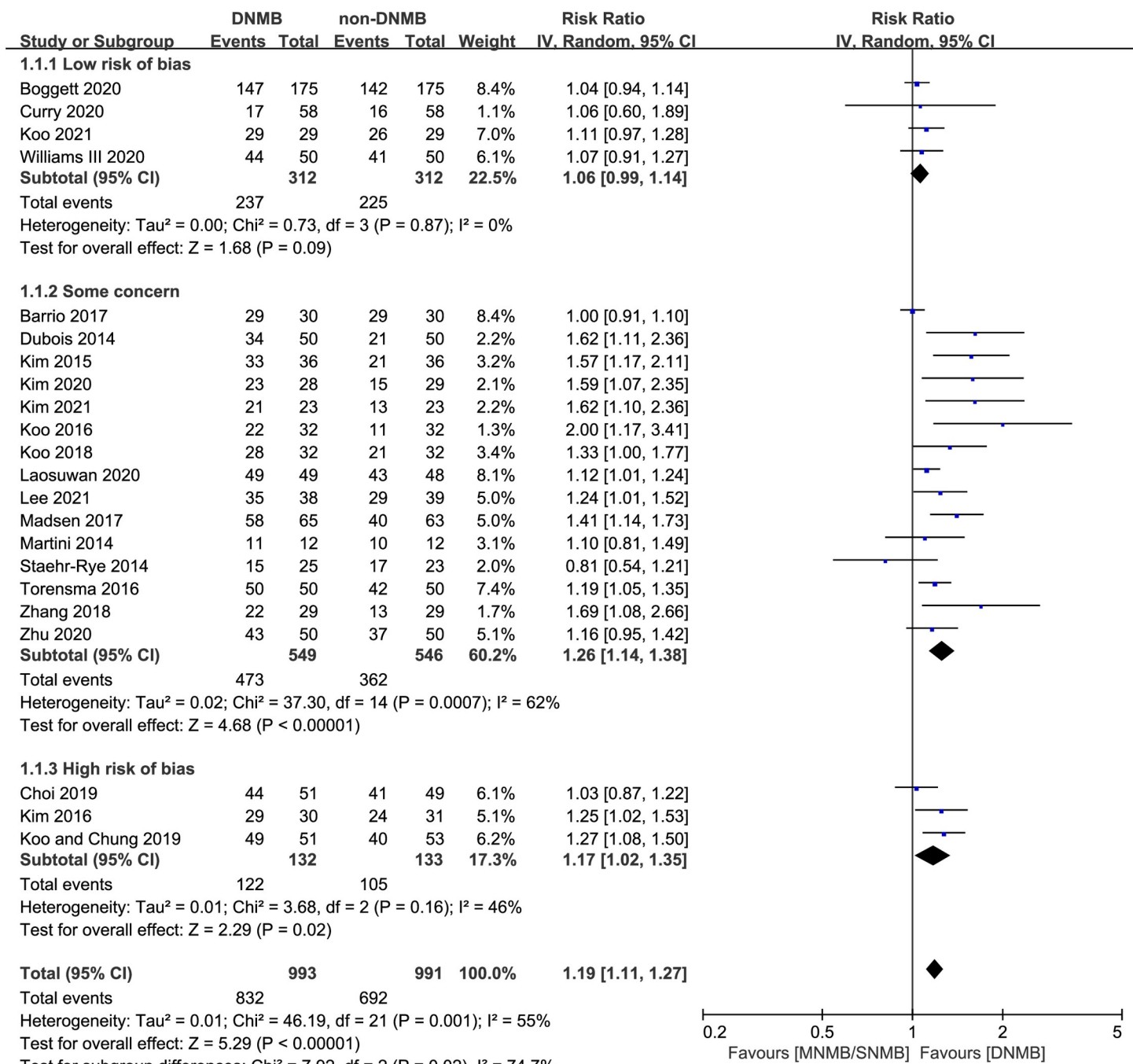

**Fig 3. Forest plot of acceptable surgical condition.**

subgroups (10 RCTs, RR: 0.71, 95% CI: [0.48, 1.04], $p$ = 0.08, $I^2$ = 59%; 2 RCTs, RR: 0.21, 95% CI: [0.04, 1.12], $p$ = 0.07, $I^2$ = 46%; 7 RCTs, RR: 0.62, 95% CI: [0.32, 1.19], $p$ = 0.15, $I^2$ = 57%) (S4 Fig).

**Intraoperative blood loss.** Five studies examined the intraoperative blood loss in 329 patients [16, 33, 35, 40, 54]. The pooled mean intraoperative blood loss was 195.7±220.9 mL in the DNMB group and 242.6±280.6 mL in the non-DNMB group. The pooled data showed that

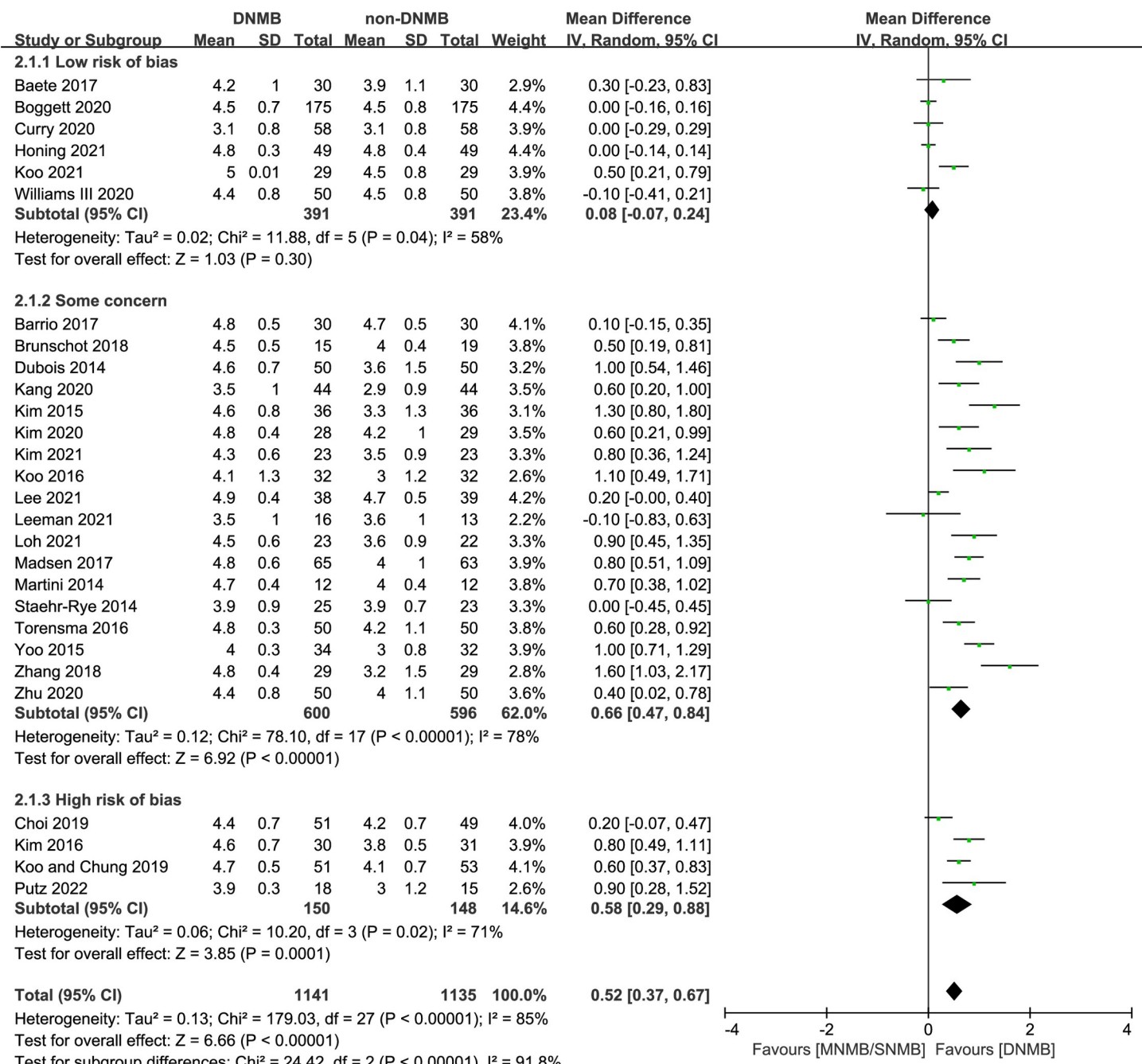

**Fig 4. Forest plot of surgical condition score.**

the difference was not significant between groups, and the heterogeneity was moderate (MD: -22.80, 95% CI: [-48.83, 3.24], $p = 0.09$, $I^2 = 51\%$) (S5 Fig).

In terms of subgroup analyses, only one study in both the orthopedic surgery subgroup and the TIVA subgroup showed that the DNMB group had lower intraoperative blood loss (1 RCT, MD: -115.00, 95% CI: [-216.90, -13.10], $p = 0.03$) (S5 Fig). No included RCT compared DNMB versus MNMB for intraoperative blood loss.

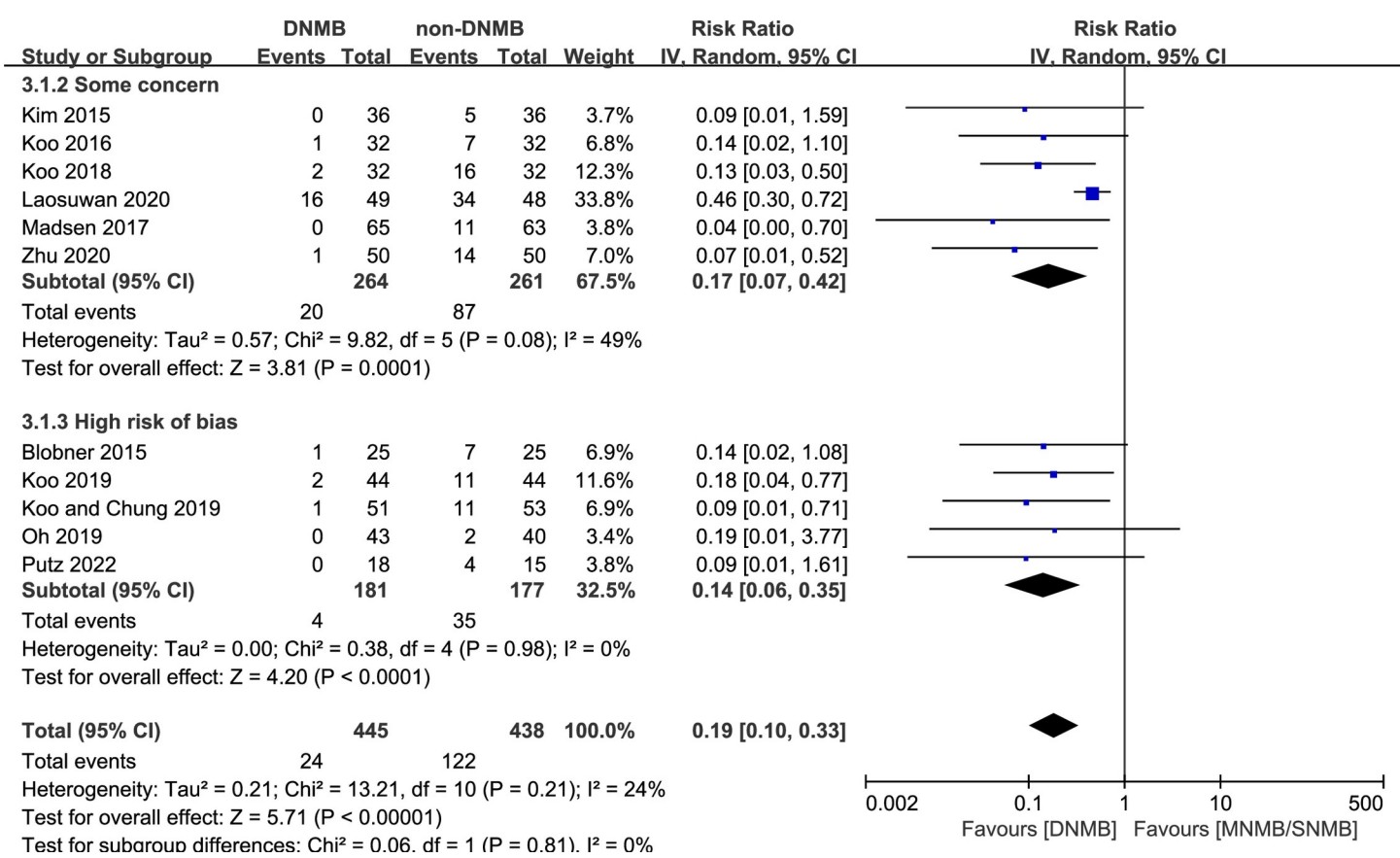

**Fig 5. Forest plot of intraoperative movement.**

**Duration of surgery.** Thirty-five studies reported the duration of surgery involving 2881 patients [12–17, 19, 23, 26, 28–33, 35–43, 45–55]. The pooled mean duration of surgery was 122.1±83.9 min in the DNMB group and 1122.5±84.5 min in the MNMB or SNMB group. The pooled data demonstrated that the difference was not significant between groups, and the heterogeneity might not be important (MD: -0.05, 95% CI: [-2.05, 1.95], $p$ = 0.96, $I^2$ = 28%). In subgroup analyses, the results were stable (S6 Fig).

**Pain at 24 h.** Ten studies reported the pain score at 24 h involving 691 patients using VA or NRS [17, 27, 30, 33–35, 45, 48, 49, 52]. The pooled mean pain score at 24 h was 2.5±1.7 in the DNMB group and 2.9±1.4 in the non-DNMB group. DNMB was found to be associated with a lower pain score at 24 h with considerable heterogeneity (MD: -0.42, 95% CI: [-0.74, -0.10], $p$ = 0.01, $I^2$ = 87%) (S7 Fig).

There was no significant difference in the orthopedic suy subgroup, non-MNMB subgroup, TIVA subgroup, or inhalational anesthesia subgroup (2 RCTs, MD: 0.00, 95% CI: [-0.19, 0.19], $p$ = 1.00, $I^2$ = 0%; 1 RCT, MD: 0.00, 95% CI: [-0.44, 0.44], $p$ = 1.00; 5 RCTs, MD: -0.29, 95% CI: [-0.60, 0.02], $p$ = 0.06, $I^2$ = 88%; 5 RCTs, MD: -0.55, 95% CI: [-1.50, 0.40], $p$ = 0.26, $I^2$ = 82%) (S7 Fig).

**Pain at 48 h.** Seven studies reported the pain score at 48 h involving 457 patients using the VAS or NRS [17, 30, 33–35, 45, 48]. The pooled mean pain score at 48 h was 1.8±1.2 in the DNMB group and 2.2±1.6 in the non-DNMB group. The pooled data found no significant difference between groups with considerable heterogeneity (MD: -0.49, 95% CI: [-1.03, 0.05], $p$ = 0.08, $I^2$ = 86%) (S8 Fig).

The difference was significant in the orthopedic surgery subgroup and non-MNMB subgroup (2 RCTs, MD: -0.71, 95% CI: [-1.20, -0.23], $p$ = 0.004, $I^2$ = 61%; 1 RCT, MD: -0.50, 95% CI: [-0.85, -0.15], $p$ = 0.005). The results were stable in the other subgroup analyses (S8 Fig).

**Length of stay.** Fourteen studies reported the length of stay involving 1312 patients [14, 15, 26, 30, 33–36, 40, 42, 45, 47, 50, 53]. The pooled mean length of stay was 4.9±4.2 in the DNMB group and 5.1±4.2 in the non-DNMB group. The pooled data showed no significant difference between groups, and the heterogeneity was not significant (MD: -0.05, 95% CI: [-0.19, 0.08], $p$ = 0.15, $I^2$ = 28%). The results were stable in all the subgroup analyses (S9 Fig).

## Sensitivity analysis and publication bias

We conducted a sensitivity analysis of all outcomes by excluding studies at high risk of bias and converting to a fixed-effect model. As a result, the pooled result on pain at 24 h was changed after eliminating 5 studies at high risk of bias (5 RCTs, MD: -0.18, 95% CI: [-0.43, 0.08], $p$ = 0.17, $I^2$ = 60%). The pooled result on intraoperative blood loss, duration of surgery, and pain at 48 h was changed after converting to a fixed-effect model (5 RCTs, MD: -9.07, 95% CI: [-17.52, -0.61], $p$ = 0.04, $I^2$ = 51%; 35 RCTs, MD: 1.12, 95% CI: [0.70, 1.55], $p$ < 0.00001, $I^2$ = 26%; 7 RCTs, MD: -0.49, 95% CI: [-0.67, -0.31], $p$ < 0.00001, $I^2$ = 86%). The other outcomes were not altered (S2 Table).

For the duration of surgery, pain at 24 h, and pain at 48 h, no significant publication bias was found through the funnel plots. For the other outcomes, the asymmetry of funnel plots was suggestive of publication bias (S10 Fig).

## Trial sequential analysis

We performed a trial sequential analysis (TSA) on the primary outcomes. TSA confirmed the effect of DNMB on acceptable surgical condition, surgical condition score, and intraoperative movement based on the type I error of 5%, the power of 80%, and the overall calculated intervention effect. The results of trial sequential analyses showed that the sample size exceeded the required information size, and the meta-analysis was robust to trial sequential analyses (S11 Fig).

## Assessment of quality of evidence

We divided the strength of evidence using the GRADE system. The quality of the pooled data for the intraoperative movement was high. The quality of the pooled data for an acceptable surgical condition and duration of surgery was moderate. The quality of the pooled data for surgical condition score, additional measures to improve surgical condition, intraoperative blood loss, and length of stay was low. Otherwise, the quality of the pooled data for pain at 24 h and pain at 48 h was very low (S3 Table).

## Discussion

This systematic review is based on 40 RCTs including a total of 3271 adult patients, which compared DNMB with shallower NMB. The pooled data suggest that DNMB improves the surgical condition (low to moderate certainty) and reduces intraoperative body movement (high certainty). DNMB is associated with a reduction in pain at 24 h (very low certainty), but it may not have clinical significance. There was no evidence of DNMB reducing intraoperative blood loss (very low certainty), duration of surgery (high certainty), pain at 48 h (very low certainty), or length of stay (moderate certainty). Furthermore, adverse events related to different depths of NMB were not reported.

In terms of the risk of bias, there were few studies at low risk of bias (7 RCTs, 17.5%) [17, 23, 26, 28, 31, 40, 53]. Some studies were judged to have some concerns about the risk of bias (23 RCTs, 57.1%) [12, 15, 16, 19, 24, 29, 32, 34, 36, 38, 41–47, 49–52, 54, 55]. There were two main sources of high risk of bias, including deviations from the intended interventions and missing outcome data. This is mainly because some small sample studies did not follow the principle of intention-to-treat analysis excluding participants after randomization without adequate explanation.

With the introduction of sugammadex, a highly selective antagonist of nondepolarizing neuromuscular blocking agents, rocuronium-sugammadex allows anesthesiologists to maintain DNMB throughout surgery and reverse DNMB immediately after surgery, which effectively reduces residual NMB [56, 57]. The purpose of this study was to investigate the effect of DNMB on the perioperative outcomes of adults undergoing general anesthesia. Both this systematic review and published reviews indicate that NMB is one of the modifiable factors affecting the surgical workspace of laparoscopic surgery, and DNMB can improve surgeons' satisfaction with the surgical condition during laparoscopic surgery [8–11].

However, it is still unclear whether DNMB has the same benefits for other types of surgery. To increase the applicability of the conclusions, non-laparoscopic surgeries (laparotomy [46], orthopedic surgery [16, 17, 28, 48], thoracoscopic surgery [14, 15], laryngeal surgery [12, 13, 41], urological surgery [39], and interventional radiography surgery [32, 45]) were included, which is one of the most important strengths of our systematic review. In addition, we performed subgroup analyses for different surgical types, and the results were consistent in terms of surgical condition score, acceptable surgical condition, intraoperative movement, duration of surgery, and length of stay but inconsistent in other outcome measurements.

In this systematic review, there were 35 studies on laparoscopic surgery, while only 5 studies were on non-laparoscopic surgery. Based on subgroup analysis, we suggested the findings were robust in laparoscopic surgery. However, due to the few studies on non-laparoscopic surgery and the small sample size, the research findings were not stable in non-laparoscopic surgery. In clinical practice, the requirements for NMB are different. Even if TOF 1–2 is detected on the adductor pollicis muscle, the diaphragm and abdominal muscle may contract because of their more resistance to NMBAs and advanced neuromuscular recovery, so laparoscopic surgery has the highest requirements for the depth of NMB [58, 59]. Orthopedic surgeons, otolaryngologists, and interventional radiologists often differ greatly in their expectations of optimal surgical conditions. Deterioration of surgical workspace conditions is manifested differently in different types of surgery, such as reduced abdominal space in laparoscopic surgery, insufficient vocal fold exposure and glottic movement in laryngoscopy surgery, and cough in thoracoscopic surgery. Briefly, the instability of subgroup analysis results across several outcomes, as well as differences in clinical practice, suggest that the type of surgery is a source of clinical and statistical heterogeneity. There are few studies on non-laparoscopic surgery, which is one of the limitations of this systematic review. Future studies with larger sample sizes and different types of surgery are needed to explore whether DNMB is important in only a subset of patients.

TOF and PTC are quantitative monitoring for NMB depth. In previous systematic reviews, TOF 1–2 is defined as MNMB, while TOF 0 and PTC 1–2 are defined as DNMB [8–11]. In a recent consensus statement on perioperative neuromuscular monitoring, DNMB is PTC > 0 with TOF = 0, and MNMB is TOF of 1–3 [7]. Among the 40 studies included in this systematic review, control groups in 8 studies [13, 25, 29, 46–48, 50, 51] did not meet the standard for the definition of specific NMB depth, in which the neuromuscular blockade was not maintained by continuous or intermittent supplemental NMBAs guided by neuromuscular monitoring, and no or smaller doses of NMBAs were used to maintain no NMB or the more superficial

and continually decreasing NMB depth. To investigate this issue, we divided the included studies into two subgroups (the DNMB vs MNMB subgroup and the DNMB vs non-MNMB subgroup). The results of subgroup analyses were stable in terms of surgical condition score, acceptable surgical condition, intraoperative movement, duration of surgery, and length of stay but inconsistent in other outcomes. Accordingly, inconsistencies in interventions may be a source of heterogeneity, while inconsistency led to a reduction in evidence certainty. Considering the large sample size in the DNMB vs MNMB subgroup, the analyses in the DNMB vs MNMB subgroup are robust. Inconsistency in neuromuscular blockade management protocol is one of the limitations. In future studies, the definition of NMB depth needs to be unified, and studies should maintain a specific depth of neuromuscular blockade by neuromuscular monitoring and report the details of the neuromuscular blockade management protocol.

In terms of outcomes, we used four outcome measures to comprehensively evaluate the impact of DNMB on surgical conditions, including surgical condition score, acceptable surgical condition, intraoperative movement, and additional measures to improve the surgical condition. Both the 5-point scale (optimal, good but not optimal, moderate, poor but not optimal, poor, and unacceptable) [19] and the 4-point scale (excellent, good but not optimal, poor but acceptable, unacceptable) [20] are used to evaluate surgical condition, which are based on the subjective judgment of the surgeon. We converted the 4-point scale to a 5-point scale. Since both scales have a similar definition (optimal, good but not optimal, poor but not acceptable, and unacceptable), conversion between different scales may not lead to considerable clinical heterogeneity [10]. It is important to note that the surgical condition score has a skewed distribution. Acceptable surgical condition outcome was used to avoid the possibility of pooling data with a skewed distribution, which may also reduce statistical heterogeneity caused by different measuring approaches. The considerable heterogeneity in the pooled results of the additional measures to improve the surgical condition may be due to the different remedies used in different studies. In laparoscopic surgery, researchers can use additional NMBAs, change body position, increase pneumoperitoneum pressure, and switch to open surgery [4]. For other types of surgery, patients only received additional NMB agents. The most commonly mentioned method to improve surgical conditions is to adjust the pneumoperitoneum pressure during laparoscopic surgery. However, there is no consensus or guideline on how to adjust pneumoperitoneum pressure, when to adjust pneumoperitoneum pressure, the range of adjustment, and whether to adjust pneumoperitoneum pressure dynamically and individually, which is worthy of further exploration in future studies.

Although the results of this systematic review directly support the improvement of surgical conditions by DNMB, there is no sufficient evidence explaining whether DNMB is associated with postoperative complications. The classical hypothesis about the relationship between DNMB and complications is that muscle contraction reduces the operation space and increases the difficulty of surgery, and sudden body movement during surgery may lead to accidental injury of important tissues and organs [3]. Boon et al. reported a significant reduction in 30-day readmission in the DNMB group compared to the shallower NMB group [60]. A retrospective study found that poorer surgical conditions were associated with more severe postoperative complications [61]. These studies indirectly support the possibility that DNMB may reduce postoperative complications. Although specific complications were reported in 21 studies in our systematic review, the sample size was small, and the physiological mechanism of DNMB affecting complications was not studied. Thus, more evidence from animal models, clinical trials, and physiological pathways is needed in the future.

This systematic review showed that the pain score was significantly reduced 24 hours after surgery. However, when we introduced the minimal clinically important difference (MCID) (1.0 for 10.0 scale) in acute pain, we considered that the pain score reduction of 0.42 may not

be clinically significant [62]. The pain score is a subjective outcome, and the postoperative analgesia plan and analgesic drug dosage should also be considered in the evaluation of post-operative pain, which are important indicators lacking in the included studies. Furthermore, Torensma et al. showed that pain reduction after bariatric surgery undergoing DNMB was independent of pneumoperitoneum pressure since all patients received the same pneumoperitoneum pressure [52].

Residual neuromuscular blockade is associated with muscle weakness, delayed emergence from anesthesia, and respiratory complications [63–65]. It is one of the most important adverse events of NMB. The incidence of residual neuromuscular blockade was 15%-89% in patients with spontaneous recovery and 3.5%-90.5% in patients with the administration of neostigmine. Sugammadex versus neostigmine further reduces the incidence of residual neuromuscular blockade, but it still occurs in 0–16% of patients [63]. All the RCTs included in this systematic review did not report the incidence of residual neuromuscular blockade, therefore the question of whether DNMB increases the incidence of residual neuromuscular blockade is beyond the scope of our systematic review. Given the importance of this issue, we suggest that future studies include the incidence of residual neuromuscular blockade as an outcome indicator.

We performed TSA analyses for the primary outcomes and found that the sample size exceeded the required information size. Therefore, the quality of evidence is not degraded by inaccuracy. Another limitation of this study was that most of the outcomes had significant publication bias, possibly because all the included studies were small and some of the studies (17/40) were funded by Merck Sharp & Dohme (MSD). We attempted to reduce publication bias by searching for unpublished and ongoing studies. Unfortunately, unpublished and ongoing studies had no available data.

In summary, we explained the statistical results, explaining the indirectness and applicability of our results from three aspects: bias risk, evidence certainty, and PICO (patient, intervention, comparison, outcome) question matching. The strength of this review is that it is designed as a systematic review, with an advanced analysis protocol and quality of evidence grade. It also adds to the first evidence of the impact of DNMB on perioperative outcomes in non-laparoscopic surgery. There are some limitations to this review. There are few studies with low bias risk, and the bias risk mainly comes from deviations from the intended interventions and missing outcome data. We note that some results have considerable heterogeneity (including surgical condition score, additional measures to improve the surgical condition, pain at 24 h, and pain at 48 h). There are few studies on adverse events. There are few studies on non-laparoscopic surgery. Some results have publication bias.

## Conclusion

In practice, DNMB improves surgical conditions and prevents intraoperative movement. There is no sufficient evidence that DNMB is associated with intraoperative blood loss, duration of surgery, complications, pain, and length of stay.

For research, future research needs to improve the quality of methodology to reduce the risk of bias. To elucidate the effect of DNMB on perioperative outcomes, more RCTs are needed, especially concerning non-laparoscopic surgery. The definition of NMB depth and outcomes needs to be unified, and it is necessary to explore the minimum clinical differences in perioperative outcome measures. More attention should be given to complications. More studies from animal models, clinical trials, and specific markers are needed to explore the physiological mechanism behind DNMB and perioperative outcomes. We will continue to update the search and include more qualified studies in the future.

## Supporting information

**S1 Fig. Forest plots of subgroup analyses showing acceptable surgical condition.** (A) Type of surgery, (B) Depth of neuromuscular blockade, (C) Type of anesthesia.
(PDF)

**S2 Fig. Forest plots of subgroup analyses showing surgical condition score.** (A) Type of surgery, (B) Depth of neuromuscular blockade, (C) Type of anesthesia.
(PDF)

**S3 Fig. Forest plots of subgroup analyses showing intraoperative movement.** (A) Type of surgery, (B) Depth of neuromuscular blockade, (C) Type of anesthesia.
(PDF)

**S4 Fig. Forest plots of analyses showing additional measures to improve the surgical condition.** (A) Primary analysis, (B) Subgroup based on the type of surgery, (C) Subgroup based on the depth of neuromuscular blockade, (D) Subgroup based on the type of anesthesia.
(PDF)

**S5 Fig. Forest plots of analyses showing intraoperative blood loss.** (A) The primary analysis, (B) Subgroup based on the type of surgery, (C) Subgroup based on the type of anesthesia.
(PDF)

**S6 Fig. Forest plots of analyses showing duration of surgery.** (A) Primary analysis, (B) Subgroup based on the type of surgery, (C) Subgroup based on the depth of neuromuscular blockade, (D) Subgroup based on the type of anesthesia.
(PDF)

**S7 Fig. Forest plots of analyses showing pain at 24 h.** (A) Primary analysis, (B) Subgroup based on the type of surgery, (C) Subgroup based on the depth of neuromuscular blockade, (D) Subgroup based on the type of anesthesia.
(PDF)

**S8 Fig. Forest plot of analysis showing pain at 48 h.** (A) Primary analysis, (B) Subgroup based on the type of surgery, (C) Subgroup based on the depth of neuromuscular blockade, (D) Subgroup based on the type of anesthesia.
(PDF)

**S9 Fig. Forest plots of analyses showing length of stay.** (A) Primary analysis, (B) Subgroup based on the type of surgery, (C) Subgroup based on the depth of neuromuscular blockade, (D) Subgroup based on the type of anesthesia.
(PDF)

**S10 Fig. Funnel plots.** (A) Acceptable surgical condition, (B) Surgical condition score, (C) Intraoperative movement, (D) Additional measure to improve the surgical condition, (E) Intraoperative blood loss, (F) Duration of surgery, (G) Pain at 24 h, (H) Pain at 48 h, (I) Length of stay.
(PDF)

**S11 Fig. Graphical results of trial sequential analyses.** (A) Acceptable surgical condition, (B) Surgical condition score, (C) Intraoperative movement.
(PDF)

**S1 Table. Search strategies for PubMed, Embase, the Cochrane Central Register of Controlled Trials (CENTRAL), and Google scholar.**
(DOCX)

**S2 Table. Sensitivity analysis.**
(DOCX)

**S3 Table. Assessment of outcome quality using the Grading of Recommendations Assessment, Development and Evaluation (GRADE).**
(PDF)

**S1 Checklist. PRISMA 2020 checklist.**
(DOCX)

## Author Contributions

**Conceptualization:** Siyuan Liu, Bin He, Lei Deng, Qiyan Li, Xiong Wang.

**Formal analysis:** Xiong Wang.

**Investigation:** Siyuan Liu, Bin He, Lei Deng.

**Methodology:** Siyuan Liu, Lei Deng, Qiyan Li, Xiong Wang.

**Project administration:** Siyuan Liu.

**Software:** Siyuan Liu, Xiong Wang.

**Supervision:** Xiong Wang.

**Validation:** Bin He, Lei Deng, Qiyan Li.

**Visualization:** Siyuan Liu.

**Writing – original draft:** Siyuan Liu, Bin He, Lei Deng, Qiyan Li.

**Writing – review & editing:** Siyuan Liu, Xiong Wang.

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
