## [Decision Letter · Decision Letter 0]

31 Oct 2022

PONE-D-22-20092Does deep neuromuscular blockade provide improved perioperative outcomes in adult patients? A systematic review and meta-analysis of randomized controlled trialsPLOS ONE

Dear Dr. Wang,

Thank you for submitting your manuscript to PLOS ONE. After careful consideration, we feel that it has merit but does not fully meet PLOS ONE’s publication criteria as it currently stands. Therefore, we invite you to submit a revised version of the manuscript that addresses the points raised during the review process.

ACADEMIC EDITOR

**Please add confidence intervals to**

**the pooled surgical score**

We look forward to receiving your revised manuscript.

Kind regards,

Silvia Fiorelli

Academic Editor

PLOS ONE

Journal Requirements:

Reviewers' comments:

Reviewer's Responses to Questions

**Comments to the Author**

1. Is the manuscript technically sound, and do the data support the conclusions?

Reviewer #1: Yes

Reviewer #2: No

2. Has the statistical analysis been performed appropriately and rigorously? 

Reviewer #1: Yes

Reviewer #2: Yes

3. Have the authors made all data underlying the findings in their manuscript fully available?

Reviewer #1: Yes

Reviewer #2: Yes

4. Is the manuscript presented in an intelligible fashion and written in standard English?

Reviewer #1: Yes

Reviewer #2: Yes

5. Review Comments to the Author

Reviewer #1: I congratulate the authors on their hard, interesting and rigorous work. The meta-analysis fully addresses the open question of the potential benefit of deep versus moderate / superficial block.

The methodology is strong and the text reads well.

Some tips to improve reading:

- Methods section: the difference between "acceptable surgical condition" and "surgical condition score" is a bit confusing. The Results and Discussion sections help to understand the difference, but the definitions in M&M need to be clear. Perhaps it would be useful to add a figure showing the surgical scale that underlines which score the authors have defined as "acceptable"

- Methods section: Similarly it is not clear what are the "measures to improve the surgical condition". Please provide more information (although the Discussion section provides some definitions)

- Results section: If possible, it would be interesting to have some data on what "measures to improve the surgical condition" have been taken

- Results section: the pooled surgical score must be reported with confidence intervals

- Discussion Section: The authors have included many types of surgeries and I agree with them that this is an important strength of their work. However, they should point out that non-laparoscopic studies are few and very heterogeneous (only 9 studies including about 700 patients). Therefore, it is true that the overall effect size is important, but the subgroup analysis is weak for non-laparoscopic studies. Large differences in optimal surgical conditions are generally expected between orthopedics, ENT surgeons, and interventional radiologists. it is difficult to target these needs and this is one of the reasons for the need for further investigation in the future.

- I have some concern about complications analysis. It is very difficult to evaluate the protective effect of NMBA reversal (both neostigmine and sugammadex). This issue is beyond the scope of the present study but it should be stated by the authors that they did not take into account in their analysis this important aspect of neuromuscular blockade management.

Reviewer #2: Unfortunately, this systematic review has a fatal flaw. It purports to compare Deep Neuromuscular Blockade (DNMB) to Moderate Neuromuscular Blockade (MNMB) or Shallow Neuromuscular Blockade (SNMB). Although they fail to use terminology that is consistent with the most recent international consensus statement (Naguib et al, Anesh Analg 2018, moderate block is TOF count of 1-3 and Shallow block is TOF count of 4 with TOF ratio < 40%), this is not the most critical issue. The most critical issus is that many of the studies the authors have included DID NOT maintain moderate or shallow block in the comparator arm. Examples are Staehr-Rye AK, et al (REF 53) in which muscle relaxation was not maintained in the 2nd arm, in Madsen et al, BJA 2017 (REF 47), the study protocol allowed the 2nd arm to have a fluctuating level of block (i.e. it was NOT consistently maintained at moderate or shallow level) and the authors admit that "a continuous moderate NMB compared with our standard NMB regimen, could have provided the same results indicating that total block of the abdominal muscles

may not be needed if patients are sufficiently anaesthetized."! In the study by Dubois et al (REF 29) Fig 3 clearly reports that surgical conditions were unsatisfactory when the block was minimal. Minimal block is now the consensus term for a block with TOF count of 4 and TOF ratio > 40%. In other words, this is clear less of a block (i.e. "shallower) than Shallow Block. This same issue applies to several of the other studies included and I will not go over the details of each one. However, clear it is meaningless to compare deep block to no block or to minimal block. It is also not adequately rigorous to compare deep block to a block of unknown depth as was done in some studies, e.g. Barrio et al, CJA 2017 (Ref 24) in which no neuromuscular monitoring was used in the non-deep arm.

The authors need to redo their analysis and include only studies in which Deep block was indeed compared to moderate (or shallow block). This requires that not only the deep blockade is maintained by infusion, guided by neuromuscular monitoring, but that the moderate (or shallow) block is also maintained by infusion, guided by neuromuscular monitoring. Without this approach, it is not possible to conclude that deep block is superior to moderate block.

6. PLOS authors have the option to publish the peer review history of their article (what does this mean?). If published, this will include your full peer review and any attached files.

Reviewer #1: No

Reviewer #2: No

---

## [Author Response · Author response to Decision Letter 0]

21 Nov 2022

Thank you for your letter and comments on our manuscript titled “Does deep neuromuscular blockade provide improved perioperative outcomes in adult patients? A systematic review and meta-analysis of randomized controlled trials” (PONE-D-22-20092). These comments helped us improve our manuscript, and provided important guidance for future research.

We have addressed the editor’s and the reviewers’ comments to the best of our abilities and revised the text to meet the PLOS ONE style requirements. We hope this meets your requirements for a publication. Relevant revisions made in response to reviewer comments, we marked the revised portions in red in the manuscript. The main comments and our specific response are detailed below:

Academic Editor: 

Please add confidence intervals to the pooled surgical score.

Response: Thanks for your suggestion, we have added confidence intervals to the pooled surgical score (Lines 265-267 in the manuscript).

Editors:

Response: Thanks for your suggestion, we have made the corresponding changes according to the submission guidelines of PLOS ONE. We have uploaded all the figure files to the Preflight Analysis and Conversion Engine (PACE) digital diagnostic tool and passed the certification. So, we re-uploaded our figure files. 

2. Upon re-submitting your revised manuscript, please upload your study’s minimal underlying data set as either Supporting Information files or to a stable, public repository and include the relevant URLs, DOIs, or accession numbers within your revised cover letter.

Response: This study is a systematic review, and all data are derived from published articles, which can be found in the manuscript and supporting information files. Our study protocol was published in the International Prospective Register of Systematic Reviews. Changed Data Availability Statement is below: A pre-specified study protocol was published in the International Prospective Register of Systematic Reviews (PROSPERO) database (CRD42022301248, https://www.crd.york.ac.uk/PROSPERO/display_record.php?RecordID=301248). All relevant data are within the manuscript and its Supporting information files.

Reviewer 1:

1. Methods section: the difference between "acceptable surgical condition" and "surgical condition score" is a bit confusing. The Results and Discussion sections help to understand the difference, but the definitions in M&M need to be clear. Perhaps it would be useful to add a figure showing the surgical scale that underlines which score the authors have defined as "acceptable". 

Response: We are sorry that this part was not clear in the original manuscript. We have added a paragraph describing definitions of outcome indicators (Lines 117-139 in the manuscript). Considering a large number of figures in the manuscript, we have not added new figures and only used words to describe the definition of "acceptable surgical condition" and "surgical condition score" (Lines 117-129 in the manuscript).

2. Methods section: Similarly it is not clear what are the "measures to improve the surgical condition". Please provide more information (although the Discussion section provides some definitions).

Response: Thanks for your suggestion, we have added a paragraph describing definitions of "additional measures to improve the surgical condition" (Lines 132-135 in the manuscript). 

3. Results section: If possible, it would be interesting to have some data on what "measures to improve the surgical condition" have been taken.

Response: Thanks for your suggestion, we have added new sentences describing specific measures to improve the surgical condition, their references, and their applicable conditions (Lines 309-315 in the manuscript).

4. Discussion Section: The authors have included many types of surgeries and I agree with them that this is an important strength of their work. However, they should point out that non-laparoscopic studies are few and very heterogeneous (only 9 studies including about 700 patients). Therefore, it is true that the overall effect size is important, but the subgroup analysis is weak for non-laparoscopic studies. Large differences in optimal surgical conditions are generally expected between orthopedics, ENT surgeons, and interventional radiologists. it is difficult to target these needs and this is one of the reasons for the need for further investigation in the future.

Response: Thanks for your affirmation and suggestion on our work. We acknowledge that the type of surgery is one of the major sources of heterogeneity. Statistically, both a small sample size and high heterogeneity suggest low evidence quality. In clinical practice, different surgeries have different requirements for the neuromuscular blockade, and different surgeries have different manifestations of deterioration of surgical workspace conditions. In the different types of surgeries, the measures to improve the surgical condition are different. Therefore, we have added a paragraph to explain this issue, and we have stated that this is one of the limitations of this paper and that this is one of the reasons for the need for further investigation (Lines 450-470 in the manuscript).

5. I have some concern about complications analysis. It is very difficult to evaluate the protective effect of NMBA reversal (both neostigmine and sugammadex). This issue is beyond the scope of the present study but it should be stated by the authors that they did not take into account in their analysis this important aspect of neuromuscular blockade management.

Response: Thanks for your suggestion. According to our pre-specified study protocol, residual neuromuscular blockade should have been one of the adverse events. We cited references to illustrate the incidence of residual neuromuscular blockade (both neostigmine and sugammadex) and its adverse consequences in the original manuscript, to demonstrate the importance and rationality of residual neuromuscular blockade as an adverse event. Unfortunately, all included studies did not report this important outcome, so the question of whether DNMB increases the incidence of residual neuromuscular blockade is beyond the scope of our systematic review. According to your suggestion, we have changed this paragraph to state this issue (Lines 541-553 in the manuscript).

Reviewer 2:

The authors need to redo their analysis and include only studies in which Deep block was indeed compared to moderate (or shallow block). This requires that not only the deep blockade is maintained by infusion, guided by neuromuscular monitoring, but that the moderate (or shallow) block is also maintained by infusion, guided by neuromuscular monitoring. Without this approach, it is not possible to conclude that deep block is superior to moderate block.

Response: Thanks for your suggestion. The details of the revised manuscript and responses are as follows:

(1) Before re-analyses, we have re-screened all the included studies and have found that 2 included studies have a fluctuating level of neuromuscular blockade in the control arm due to their cross-over design (Madsen 2015-REF 47, Söderström 2018-REF 52 in the original manuscript) and 8 included studies have no specific level of neuromuscular blockade in the control arm (REF 13, 25, 29, 46, 47, 48, 50, 51 in the revised manuscript). Based on the pre-specified study protocol, we should only include parallel randomized control trials, so we have removed two cross-over studies that have a fluctuating level of neuromuscular blockade in the control arm (Madsen 2015-REF 47, Söderström 2018-REF 52 in the original manuscript). 

(2) According to our pre-specified study protocol (CRD42022301248), the intervention is the deep neuromuscular block and the comparator is the shallower neuromuscular block (from Moderate block to acceptable recovery). We are sorry that we used the unclear word “MNMB/SNMB group” to describe the shallower neuromuscular block. We have changed this issue and used “DNMB group” and “non-DNMB group” to describe the interventions. (Lines 106-107 in the manuscript). 

(3) We do not think it is meaningless to include the shallower blockade than MNMB (including shallow block, minimal block, and acceptable recovery) in the control arm. Although, from your point, “in the study by Dubois et al (REF 29) Fig 3 clearly reports that surgical conditions were unsatisfactory when the block was minimal”, Dubois et al only included participants receiving laparoscopic surgery and did not report other outcomes (including intraoperative movement, and adverse events et al.). As we all know, different surgeries have different requirements for the level of neuromuscular blockade, while orthopedic surgeons, otolaryngologists, and interventional radiologists often differ greatly in their expectations of optimal surgical conditions. The loose inclusion criteria are one of the main strengths of this systematic review, designed to increase the applicability of the conclusions. 

In the revised manuscript, we have reserved 8 included studies with no specific level of neuromuscular blockade in the control arm (REF 13, 25, 29, 46, 47, 48, 50, 51 in the manuscript), and we have used “non-MNMB” to describe the constantly changing level of neuromuscular blockade. To adequately and rigorously compare DNMB and MNMB, we have divided studies into two subgroups (the DNMB vs MNMB subgroup and the DNMB vs non-MNMB subgroup). In the DNMB vs non-MNMB subgroup, moderate neuromuscular blockade was maintained and guided by neuromuscular monitoring. After re-analyses, we found that the main conclusions had not changed. Considering the large sample size in the DNMB vs MNMB subgroup, the analyses in the DNMB vs MNMB subgroup are stable. We have changed the paragraph in the discussion part to adequately and rigorously explain this issue (Lines 471-490 in the manuscript).

(4) According to the results of re-analyses, we have revised all figures, tables, and supporting information files.

---

## [Decision Letter · Decision Letter 1]

5 Jan 2023

PONE-D-22-20092R1Does deep neuromuscular blockade provide improved perioperative outcomes in adult patients? A systematic review and meta-analysis of randomized controlled trialsPLOS ONE

Dear Dr. Wang,

Thank you for submitting your manuscript to PLOS ONE. After careful consideration, we feel that it has merit but does not fully meet PLOS ONE’s publication criteria as it currently stands. Therefore, we invite you to submit a revised version of the manuscript that addresses the points raised during the review process.

ACADEMIC EDITOR: please assess all the reviewers comments

We look forward to receiving your revised manuscript.

Kind regards,

Silvia Fiorelli

Academic Editor

PLOS ONE

Journal Requirements:

Reviewers' comments:

Reviewer's Responses to Questions

**Comments to the Author**

1. If the authors have adequately addressed your comments raised in a previous round of review and you feel that this manuscript is now acceptable for publication, you may indicate that here to bypass the “Comments to the Author” section, enter your conflict of interest statement in the “Confidential to Editor” section, and submit your "Accept" recommendation.

Reviewer #1: All comments have been addressed

Reviewer #2: (No Response)

2. Is the manuscript technically sound, and do the data support the conclusions?

Reviewer #1: Yes

Reviewer #2: Yes

3. Has the statistical analysis been performed appropriately and rigorously? 

Reviewer #1: Yes

Reviewer #2: Yes

4. Have the authors made all data underlying the findings in their manuscript fully available?

Reviewer #1: Yes

Reviewer #2: Yes

5. Is the manuscript presented in an intelligible fashion and written in standard English?

Reviewer #1: Yes

Reviewer #2: Yes

6. Review Comments to the Author

Reviewer #1: (No Response)

Reviewer #2: I wish to congratulate the authors on a greatly improved manuscript. I believe that this systematic review is a meaningful addition to existing literature on the subject. However, it is extremely important to emphasize the significant limitations in the existing literature. Systematic reviews are limited by the quality of the individual trials analyzed and reviewed. What may be viewed as “minor” variations in methodology may markedly affect outcome. Some frequently cited studies are often poorly designed to give us the answers we really need.

I assume that the authors agree that what is needed are studies that help us understand how to manage perioperative muscle relaxation for optimal patient outcomes. From this perspective, it is troubling that this area of research has greatly influenced by industry (Merck, MSD) who may have their own reasons for being involved with this line of research. The magnitude of industry involvement needs to be transparently disclosed, please add data on which studies were supported by Merck, this can be done with a simple table.

The manuscript is importantly improved with the separation of comparative studies in two categories, DNMB and non-DNMB, respectively. However, it should be stated as a limitation that we do not know in several studies how well controlled the moderate block was. Was it indeed consistently a moderate block?

Your manuscript cites a review by Richebe which is generally supportive of the concept of DNMB. I suggest you mention that the topic is controversial and you may want to cite a critical review, too. You may consider Naguib and Kopmans critical review of the topic (Anesth Analg 2015;120:51–8).

Your manuscript also does not discuss a different and potentially clinically important conclusion which is that DNMB may be important only in a subset of patients. It is possible that this subset of patients could be identified by improved intraoperative communication between surgeon and anesthesiologist.

One important aspect is whether opportunities are sought to lower the pneumoperitoneal pressure. How often is this done? Should future studies address this? Many institutions and surgeons use the same pressure for all patients, is this rational? All of these issues amount to shortcomings in the current literature and they limit what we can conclude from systematic reviews. Nevertheless, I believe the authors’ systematic review is one of the best that has been presented so far.

7. PLOS authors have the option to publish the peer review history of their article (what does this mean?). If published, this will include your full peer review and any attached files.

Reviewer #1: No

Reviewer #2: No

---

## [Author Response · Author response to Decision Letter 1]

19 Jan 2023

Dear editors and reviewers:

Thank you for your comments and suggestions on our manuscript titled “Does deep neuromuscular blockade provide improved perioperative outcomes in adult patients? A systematic review and meta-analysis of randomized controlled trials” (PONE-D-22-20092R1). These helped us improve our manuscript.

We have addressed the issues to the best of our abilities and marked the revised portions in red in the manuscript. We hope these meet your requirements for a publication. The main comments and our specific response are detailed below:

Editors:

1. However, it is extremely important to emphasize the significant limitations in the existing literature. Systematic reviews are limited by the quality of the individual trials analyzed and reviewed. What may be viewed as “minor” variations in methodology may markedly affect outcome. Some frequently cited studies are often poorly designed to give us the answers we really need.

Response: Thanks for your comment. We agree with you. Different articles are bound to differ in methodology. The main role of a systematic review is to comprehensively explore the similarities and differences between different articles and cautiously draw conclusions. In this systematic review, the quality of each article was evaluated by the risk of bias assessment (lines 238-245 in the Revised Manuscript with Track Changes), and the quality of evidence for each conclusion was evaluated accordingly by the Grades of Recommendation, Assessment, Development and Evaluation (GRADE) system (lines 406-413). Consequentially, conclusions based on higher-quality articles have higher evidence quality. According to the limitations of existing articles, this systematic review puts forward suggestions for improvement in the conclusion part, which is conducive to improving the quality of future research. We believe that with the improvement of the quality of the research design, the quality of conclusions will be improved.

2. The magnitude of industry involvement needs to be transparently disclosed, please add data on which studies were supported by Merck, this can be done with a simple table.

Response: Thanks for your suggestion. We had a statement about this limitation (lines 571-574). In reference to your suggestion, we have added the funding source item in Table 1.

3. The manuscript is importantly improved with the separation of comparative studies in two categories, DNMB and non-DNMB, respectively. However, it should be stated as a limitation that we do not know in several studies how well controlled the moderate block was. Was it indeed consistently a moderate block?

Response: Thanks for your comment. In the primary analysis, we divided the interventions into two groups (DNMB versus non-DNMB). Non-DNMB is the more superficial level of a neuromuscular blockade than DNMB, including moderate neuromuscular blockade, shallow block, and minimal block (lines 61-66). There was a limitation that the neuromuscular blockade management protocols are different among studies, and some studies did not provide the details of the depth of neuromuscular blockade or did not use neuromuscular monitoring to maintain a specific depth of neuromuscular blockade (Table 1). To address this issue, we divided the non-DNMB group into two subgroups including DNMB vs MNMB subgroup and the DNMB vs non-MNMB subgroup (lines 182-186). We have stated this limitation and suggested future studies should maintain a specific depth of neuromuscular blockade by neuromuscular monitoring and report the details of the neuromuscular blockade management protocol (lines 494-500).

4. Your manuscript cites a review by Richebe which is generally supportive of the concept of DNMB. I suggest you mention that the topic is controversial and you may want to cite a critical review, too. You may consider Naguib and Kopmans critical review of the topic (Anesth Analg 2015;120:51–8).

Response: Thanks for your suggestion. We have added a statement (lines 56-57) and cited the critical review you mentioned to emphasize that the topic is controversial (Ref 3).

5. Your manuscript also does not discuss a different and potentially clinically important conclusion which is that DNMB may be important only in a subset of patients. It is possible that this subset of patients could be identified by improved intraoperative communication between surgeon and anesthesiologist.

Response: Thanks for your comment. We agree with you. Our systematic review attempted to address this issue. In the study protocol, we hypothesized that different types of surgery may have an effect on the outcome, which is one of the important strengths of our manuscript compared with other systematic reviews (line 183). In the subgroup analysis, we divided the type of surgery into multiple subgroups, including laparoscopic surgery, laparotomy, orthopedic surgery, thoracoscopic surgery, laryngeal surgery, urologic surgery, and interventional radiography. The results showed that the outcome was stable in the laparoscopic surgery subgroup. In other surgical subgroups, the heterogeneity was high, and the sample size was small, so the outcome was unstable. We think that we draw a reasonable conclusion based on existing articles. Future studies with larger sample sizes and different types of surgery are needed to test the hypothesis that DNMB may be important only in a subset of patients. We had a paragraph in the discussion section to emphasize this limitation (lines 473-475).

6. One important aspect is whether opportunities are sought to lower the pneumoperitoneal pressure. How often is this done? Should future studies address this? Many institutions and surgeons use the same pressure for all patients, is this rational? All of these issues amount to shortcomings in the current literature and they limit what we can conclude from systematic reviews.

Response: Thanks for your comment. We think the problem you mentioned is worth exploring. There is no unified consensus or guideline on how to manage pneumoperitoneum pressure, and the management plan of each hospital is not the same. Whether DNMB can improve perioperative outcomes and its mechanism remain unclear, and one of the mechanisms may be DNMB allowing lower pneumoperitoneum pressure. Based on the existing studies, we cannot solve the questions you mentioned, and these questions are not able to be answered by this systematic review. Considering the important value of the questions you mentioned, we think it is necessary to emphasize this limitation of current studies in the discussion section. So, we added a statement. We expect more and more researchers to pay attention to these issues (lines 522-528) and hope that these issues can be solved in future studies and systematic reviews. 

We would be glad to respond to any further questions and comments that you may have.

Sincerely yours,

Wang Xiong

---

## [Decision Letter · Decision Letter 2]

23 Feb 2023

Does deep neuromuscular blockade provide improved perioperative outcomes in adult patients? A systematic review and meta-analysis of randomized controlled trials

PONE-D-22-20092R2

Dear Dr. Wang,

We’re pleased to inform you that your manuscript has been judged scientifically suitable for publication and will be formally accepted for publication once it meets all outstanding technical requirements.

Kind regards,

Silvia Fiorelli

Academic Editor

PLOS ONE

Additional Editor Comments (optional):

Congratulations to the authors and thanks to the reviewers for the suggestions provided which really helped improve the quality of the manuscript.

Reviewers' comments:

Reviewer's Responses to Questions

**Comments to the Author**

1. If the authors have adequately addressed your comments raised in a previous round of review and you feel that this manuscript is now acceptable for publication, you may indicate that here to bypass the “Comments to the Author” section, enter your conflict of interest statement in the “Confidential to Editor” section, and submit your "Accept" recommendation.

Reviewer #1: All comments have been addressed

Reviewer #2: All comments have been addressed

2. Is the manuscript technically sound, and do the data support the conclusions?

Reviewer #1: Yes

Reviewer #2: Yes

3. Has the statistical analysis been performed appropriately and rigorously? 

Reviewer #1: Yes

Reviewer #2: Yes

4. Have the authors made all data underlying the findings in their manuscript fully available?

Reviewer #1: Yes

Reviewer #2: Yes

5. Is the manuscript presented in an intelligible fashion and written in standard English?

Reviewer #1: Yes

Reviewer #2: Yes

6. Review Comments to the Author

Reviewer #1: Dear Authors,

my congratulations for the hard work to write and revise the manuscript. The R2 version reads well.

Reviewer #2: (No Response)

7. PLOS authors have the option to publish the peer review history of their article (what does this mean?). If published, this will include your full peer review and any attached files.

Reviewer #1: No

Reviewer #2: No

---

## [Editor Report · Acceptance letter]

27 Feb 2023

PONE-D-22-20092R2 

Does deep neuromuscular blockade provide improved perioperative outcomes in adult patients? A systematic review and meta-analysis of randomized controlled trials 

Dear Dr. Wang:

I'm pleased to inform you that your manuscript has been deemed suitable for publication in PLOS ONE. Congratulations! Your manuscript is now with our production department. 

Kind regards, 

on behalf of

Dr. Silvia Fiorelli 

Academic Editor

PLOS ONE